# A Systematic Review on Cloud Storage Mechanisms Concerning e-Healthcare Systems

**DOI:** 10.3390/s20185392

**Published:** 2020-09-21

**Authors:** Adnan Tahir, Fei Chen, Habib Ullah Khan, Zhong Ming, Arshad Ahmad, Shah Nazir, Muhammad Shafiq

**Affiliations:** 1Research Institute of Network and Information Security, Shenzhen University, Shenzhen 518060, China; adnantahir@szu.edu.cn (A.T.); fchen@szu.edu.cn (F.C.); 2College of Computer Science and Software Engineering, Shenzhen University, Shenzhen 518060, China; mingz@szu.edu.cn; 3Department of Accounting and Information System, College of Business and Economics, Qatar University, Doha P.O. Box. 2713, Qatar; 4Department of Computer Science, City University of Science and Information Technology, Peshawar 25000, Pakistan; yaarshad@gmail.com; 5Department of Computer Science, University of Swabi, Ambar 23430, Pakistan; shahnazir@uoswabi.edu.pk; 6Cyberspace Institute of Technology, Guangzhou University, Guangzhou 510006, China; srsshafiq@gmail.com

**Keywords:** cloud computing, cloud storage, availability, e-Healthcare, replication, erasure coding, deduplication

## Abstract

As the expenses of medical care administrations rise and medical services experts are becoming rare, it is up to medical services organizations and institutes to consider the implementation of medical Health Information Technology (HIT) innovation frameworks. HIT permits health associations to smooth out their considerable cycles and offer types of assistance in a more productive and financially savvy way. With the rise of Cloud Storage Computing (CSC), an enormous number of associations and undertakings have moved their healthcare data sources to distributed storage. As the information can be mentioned whenever universally, the accessibility of information becomes an urgent need. Nonetheless, outages in cloud storage essentially influence the accessibility level. Like the other basic variables of cloud storage (e.g., reliability quality, performance, security, and protection), availability also directly impacts the data in cloud storage for e-Healthcare systems. In this paper, we systematically review cloud storage mechanisms concerning the healthcare environment. Additionally, in this paper, the state-of-the-art cloud storage mechanisms are critically reviewed for e-Healthcare systems based on their characteristics. In short, this paper summarizes existing literature based on cloud storage and its impact on healthcare, and it likewise helps researchers, medical specialists, and organizations with a solid foundation for future studies in the healthcare environment.

## 1. Introduction

Before the advent of cloud computing, data used to be stored and retrieved locally from a single machine, but the single machine was not reliable due to synchronization issues of the backup data. In case of damage, the users were not able to retrieve recent data from that local machine. Moreover, keeping the data secure in that single machine was another issue. The synchronization of the data in the distributed system was also crucial. Moreover, the limitations of storage for a single machine was another issue in case the volume of data exceeded the machine’s storage limit.

Cloud computing is defined as a network of distributed computing on a large scale which computes through highly available, dynamically configurable/reconfigurable, and scalable resources. According to the definition of cloud computing offered by the National Institute of Standard and Technology (NIST) [1], it is a self-service to a user which is provided on-demand; can have access to a broad network where elasticity and scalability of resources are rapid; provides pooling of resources at a multi-tenant level; and finally services can be measured through a manageable, monitored, and controlled transparency system. Cloud computing does not only provide the on-demand services but also offers high availability, reliability, vast scalability, and a cheaper environment for computing to the users. Due to this nature of cloud computing, most of the organizations and industries are transforming and shifting their IT structure or model towards the cloud. The reason for this major shift is because of the advantages that cloud computing provides like the capabilities of provision computing, etc. Scaling networks through credit cards and bags filled with a bunch of tasks are some of the promises made by clouds [2].

Over time, it has been observed that the amount of data is increasing exponentially. Large and huge datasets of scientific domains are also becoming much important to the shared resources. Usually, these kinds of huge and massive data or datasets are stored in data centers residing in cloud. As most of the scientific, as well as organization’s and industry’s data, is in large volume, i.e., in Terabytes (TB) or Petabytes (PB) [3], the space for storage, resources for their processing, power for computation, and their maintenance will not necessarily be available to organization or industry. Therefore, most of the companies have utilized the services of cloud in this regard in which they do not have to keep such a huge storage and computation resources in-house.

Distributed storage assumes a significant function in the current computerized time because of the noteworthy headway of medical care advancements. As the wellsprings of cloud storage worried in healthcare organizations and different segments are notable for their volume and decent variety, thus, the healthcare services are picked up its impact through the effect of cloud storage. The healthcare organizations have produced a huge measure of medical services information over recent years. Medical services information for the most part consolidates medical records electronically known as EMRs, for example, keeping the history of patient’s medical health and clinical data, doctor notes, reports generated at the clinical centers, biometric information, and other clinical information identified with healthcare systems [4]. All this information together outcome in the cloud storage of healthcare services. The development of these services in cloud storage is advance and savvy for both open and private medical care. The accomplishment of medical care applications concerning capacity in cloud completely depends upon the basic engineering and utilization of reasonable apparatuses as demonstrated in spearheading research endeavors. It additionally gives a thought of the analysis of cloud storage in healthcare frameworks. All the more explicitly, cloud storage investigative devices and methods can improve the nature of clinical administrations and decrease the clinical expense of patients by investigating the affiliation and understanding the idea of medical services information. In 2016, Kohli et al. [5] examine how health records electronically encourage reconciliation of patient medical history for arranging protected and appropriate treatment.

The availability of data is the main concern in cloud storage. By availability, it is meant that the data should be available and accessible whenever it is requested or needed by the users. As the volume of data is increasing every day, the availability and security of data are becoming of more concern to the cloud storage providers. Therefore, more efficient methods and optimization techniques are needed as simply increasing the replicas or copies of data will also demand the increment in the storage space and cost factor. However, keeping data available at full time in cloud storage is much more necessary; otherwise, immeasurable loss could occur.

### 1.1. Problem Definition

The domain knowledge and literature show that enormous research is present and is accounted for by numerous scientists and researchers around the globe for healthcare and medical services in cloud storage. This exploration centers around separating noteworthy highlights of cloud storage in the healthcare medical services uses of medical care in cloud storage, and cutting-edge methods proposed in the cloud storage and medical services cloud storage field which can, at last, be utilized for dynamic in health and medical care. The present work sorts and sums up the current distributed exploration work dependent on the formulated research questions and identified keywords recognized for the searching cycle. The analysis in the present research work will support the specialists, clinical specialists, general doctors, and professionals to make more reliable conclusions, which eventually will assist with utilizing the investigation as proof for treating patients and propose medicines likewise.

### 1.2. Contributions

Based on the criterion defined, i.e., inclusion, exclusion, and quality assessment, a total of 90 most primary applicable studies were included in this paper. The main contributions made are as follows.

To highlight the research work in the field of cloud storage mechanisms and its impact on healthcare environment did from the year 2007 till 2020To briefly present a summary of the techniques used for cloud storage and healthcare in cloud storageTo highlight the benefits of healthcare in the field of cloud storage

The paper is organized as follows. The conducting of a detailed process of research with a systematic literature review based on guidelines is presented in Section 2. Cloud-based healthcare challenges and requirements are discussed in Section 3, and considerable factors of cloud storage and their benefits are discussed in Section 4. Basic Cloud storage mechanisms, state-of-the-art storage mechanisms, and cloud-based healthcare mechanisms are presented in Section 5. Section 6 and Section 8, Section 9 and Section 10 represent the limitations and conclusion of the current research work.

## 2. Strategic Systematic Literature Review (SSLR) Workflow

The following objectives are focused on the research process for conducting a Strategic Systematic Literature Review (SSLR) in this paper.

Evaluating diverse aspects on the idea of distributed cloud storage in the health and medical care context.Exploring the sources of health and medical care in distributed cloud storageInducing concentration to overcome the challenges of cloud storage in health and medical care

Through the in-depth discussion of these goals, the SSLR aims to help with understanding the general effect of distributed cloud storage and its applications in the health and medical care domain.

### 2.1. Research Questions

The main Research Questions (RQ) that are formulated to address and conduct the SSLR of the proposed study are as follows.

**RQ1.** What are the features of cloud storage in the medical healthcare field?**RQ2.** What are the challenges and opportunities of healthcare in cloud storage?**RQ3.** What methods and frameworks are used for cloud storage in healthcare systems?**RQ4.** What are the applications of cloud storage in healthcare?**RQ5.** What research has been published in cloud storage mechanisms and healthcare cloud storage since 2007?

### 2.2. Search Strategy

To look for the acquisition of relevant articles for our SSLR, we have used six main electronic research repositories: IEEE Xplore, ACM, Taylor and Francis, Science Direct, Wiley Online Library, and Springer. However, some of the works published by MDPI and Hindawi, which are fairly relevant to our domain, are also included in this study.

### 2.3. Search Keywords

For the searching process, we defined the following keywords; “cloud availability”, “cloud storage”, “healthcare cloud storage”, and “e-Healthcare cloud storage” in the context of the research domain. From the table of content for these keywords, we carried out an automatic in-depth text search by search engine screening and manual screening. To conduct an SSLR, Boolean operators were used with the predefined keywords and within the scope of our formulated research questions to classify relevant articles. We applied a specified time duration limit on our search according to the inclusion and exclusion criteria, so that all relevant papers should be identified and collected having a focus on the predefined keywords. To make the identified research papers to a manageable and reduced size, the paragraphs of these papers were further scanned out with the keywords.

### 2.4. Selection Criteria

Based on the following inclusion–exclusion criteria, we screened, filtered, and sorted the article in this paper: 


*Inclusion Criteria*


The articles relevancy to the cloud storage mechanisms and healthcare in cloud storageThe articles published during the years 2007–2020English should be the main and primary languageSelection of only primary studies from the relevant researchThe article should provide a sound knowledge from the research questions formulated


*Exclusion Criteria*


The articles before 2007Less than three pages of research articlesGray PapersDuplicate version papersProvide no information for the research question formulated

### 2.5. Study Selection Process

We performed the SSLR process in four stages. Figure 1 depicts the details of the study selection we carried out for the SSLR process. In the initial phase, after screening, as per the searching keywords all the articles were selected relevant to cloud storage, cloud availability, healthcare, and cloud-based healthcare storage. Then, articles were further screened according to the designed inclusion–exclusion criteria and irrelevant articles were not selected and excluded based on exclusion criteria. To be more clear that we included all the relevant and focused articles in our literature review and analysis in the first phase, we went through the bibliographies of these finally identified articles. In the second phase of SSLR screening, articles were collected based on their keywords string, title, and abstract. These articles were excluded which were not having an association with the proposed study. Now, these filtered papers were selected for the in-depth reading and analysis which included at least one of the keywords for more than 3 times in their literature content. Finally, in the last stage of screening, Boolean AND operator were used along with the predefined keywords to further filter the paper based on abstract. As an outcome, the entire screening process from the initial to the last phase helped us to identify the most relevant papers and reduce the sample size from 1430 papers to 90 papers, referring to further study by the authors.

The outcome of the in-depth reading and analysis resulted in the following schematic classification and dimensionality; (1) theme of Availability in Cloud Storage, (2) purpose of application, (3) level and characteristics of Availability in cloud storage, (4) mechanisms of availability, and (5) finally the reliability in cloud storage.

### 2.6. Quality Assessment

Quality assessment plays an important and essential role in the SSLR protocol during the review process. All the articles were quality assessed, by all authors, after the analysis and evaluation of abstracts of selected articles. During the reviewing and screening process, quality valuation assumes a huge part in the SSLR procedure. The investigation and assessment of the abstract of selected articles were performed by all the authors to evaluate the quality of finalized and screened articles. These articles, based on the inclusion–exclusion criteria, were selected for each of the formulated research questions.

### 2.7. Results and Discussion

During the reviewing and screening process of SSLR in the present research article, articles were collected based on predefined searching keywords through a search operation on the six most common electronic repositories: ACM, IEEE Xplore, Science Direct, Springer, Taylor and Francis and Wiley. In the preliminary phase, approximately 5163 articles were collected for the years 2007–2020. In the next phase, screening on the basis of title and keywords filtered out total of a 1430 articles. In the final phase of the SSLR process, the Boolean AND operator were used within the predefined searching keywords to filter the remaining article based on the abstract. The resulting 90 articles, according to each defined research question, were finalized and selected for further study and analysis by the authors according to the inclusion–exclusion criteria.

A scientometric analysis, similar to the one as represented by Sabine et al. In [6], is also carried out before for addressing the results and analysis of the content of our literature review study. The focus of this scientometric analysis is on the research methodologies used and their year of publication. As can be depicted in Table 1, there has been an increase in the number of publications since 2007 which has reached its peak in 2013. From Table 1 it can also be seen that there has been a new momentum is gained by the cloud storage availability in 2016 with 10 articles after the decline in the number of publications in 2015. Subjecting specifically to our literature review of more focused 90 research articles, Table 1 follows a classification scheme that concerns the categorization of research methods been used in these articles. Research methods that are been used for the availability of cloud storage over the 10+ years has been distributed in Table 1 with majority of them focusing on the conceptual nature (15 papers), while others concluding results from case study work (25 papers). Research methodologies that have been rarely used in the current research are action research (13 papers), ground theory (8 papers), field experiment (6 papers), and archival research (9 papers). Whereas in contrast to all these above methods, the research methods like lab experiments (7 papers) and field studies (14 papers) helped in gaining the concept of cloud storage availability and represents one-third of our total literature.

## 3. Cloud-Based Healthcare Challenges and Requirements

Although the healthcare industry enjoys valuable benefits from the cloud-based healthcare architecture, but unfortunately, the major challenges of healthcare technology and cloud computing together remain in existence which adds more weight to requirements and encounters for storing and processing sensitive medical data. In this section, we summarize some of the technical and non-technical requirements and challenges faced by the healthcare cloud.

### 3.1. Technical

**Availability:** Most clinical consideration providers require high availability and accessibility of cloud-based health and medical care organizations. Availability and data accessibility are basic requirements for clinical administration providers who cannot effectively work unless their applications and patients’ data are not available. The cloud-based health and medical care organizations should be available reliably with no impedance or data corruption. Cloud organizations could experience frustrations due to programming and gear inadequacies, arrange disappointments, security dangers, and disastrous occasions among various reasons. As CSC assets are scattered over an open cloud infrastructure, for instance, the Internet, they will not offer better accessibility stood out from guaranteeing and keeping up IT structures inside the affiliation.

**Reliability of Data and Service:** Using the cloud for a noteworthy application like cloud-based health and medical care require affirmations of good reliability for the offered kinds of administrations. All cloud-based medical services organizations and data must be with no failures. Some critical decisions regarding single human or society healthcare administrations can be taken depending upon the data and organizations given by the cloud-based health and medical care. As such organizations are passed on and may start from different Cloud providers, the chance of having imperfect or wrong data, or organizations, can increase. The data in cloud-based health and medical care must be dependable and ceaselessly in a legitimate state paying little intellect to any item, product, or framework failures. Additionally, all cloud-based health and medical care must pass on fault-tolerant organizations for clinical administration providers.

**Data Management:** Huge amounts of clinical and medical records and images related to an enormous number of people will be taken care of in cloud-based medical care. The data may be replicated for high trustworthiness and better access to different regions and across tremendous geographic divisions. A segment of the data could be moreover made open locally. Most health and medical applications require secure, gainful, strong, and adaptable permission to the clinical records. These necessities actualize the need to have some cloud storage organizations that offer transformation to intellectual faults, secure limit over open fogs, and rich inquiry accents that grant viable and flexible workplaces to recover and deal with the application data.

**Flexibility and Scalability:** A cloud-based healthcare system must be equipped for serving various health and medical care providers with various necessities. These prerequisites are as far as capacities, activities, clients, evaluating, the board, and quality of services (QoS) requirements. The cloud-based medical care frameworks and administrations should be adaptable enough to be arranged for various medical services providers’ necessities. Similarly, the e-Health Cloud should be genuinely versatile in adding new expected organizations to help clinical administration measures. While e-Health Cloud organizations must be versatile to meet differing clinical administration essentials, they moreover ought to be viably configurable to meet with different needs prerequisites. Therefore, the structure of cloud administrations to meet different requirements must be cultivated with the least effort and cost.

**Maintainability:** Dissimilar to having an e-Healthcare framework for singular medical care specialists and providers, a cloud-based healthcare system can be utilized for many medical care organizations. This expands the multifaceted nature of framework practicality in the cloud-based healthcare contrasted with a singular healthcare framework. The extension is for the most part a result of the need to consider the requirements and characteristics of the diverse healthcare organizations providers and clients. These necessities can be uncommon while up-keeping in the cloud systems, programming, or stages must be overseen without having any negative consequences for any organization obliged with any client.

### 3.2. Non-Technical

**Change in an Organization:** The shifting from in-house towards cloud-based healthcare infrastructure will require huge changes to clinical, medical, and business communities and to the authoritative limits in the medical services industry. This analysis is involved with the progressions that a cloud-based healthcare will present upon members.

**Data Ownership:** Responsibility in the medical care industry by and large is a territory with no evident rules and regulations. Can a doctor guarantee possession for the patient’s records which could be the sole property of the patient? Should there be some back-up plan for the patient’s back-up plan or the emergency clinic executives? This challenge is involved in the development of organizations and policies that draw clear ownership limits.

**Issue with Liability, Trust, and Privacy:** This requirement and challenge relates to the prospects of exposure to private and sensitive information, information loss, and the dearth of information about the area and purview of the clinical information. From the point of medical services providers’, cloud-based healthcare presents a high exposure of commitment and legal liabilities in instances of information loss or threat that may cause harm to reputation and patients’ trust.

**Consumer Usability Experience:** This requirement and difficulty regards the degree and level of acquisition received by the cloud-based healthcare clients including patients, medical services experts, and regulatory and protection faculty.

## 4. Cloud Storage Factors of Consideration

There are four main basic factors of cloud storage availability that needs to be considered when talking about cloud data storage. A general overview of these factors is depicted in Figure 2. The following is a detailed discussion of these factors.

### 4.1. Vendor Lock-In

Features like data distribution over geo-locations, specific Application Program Interfaces (APIs), etc. are some that are being offered by most of today’s Cloud Storage Providers. Users may move from one provider to another if the new provider offers services that are more attractive and cost-effective or the old providers may increase the pricing of its offered services. Somehow, the switching cost of moving from one provider to another may be very high [7] as it is directly proportional to the amount of data stored in the original cloud. Therefore, it may be very expensive for one user to move to any other cloud if the data stored in the original cloud is in large quantity. Taking this as an advantage, most of the storage providers may increase their prices or makes new contracts that may or may not be favorable to the stored data owner. Therefore, this creates a situation for the cloud storage user to only accept the changes made by their original cloud instead of moving to others and is known as the *“vendor lock-in”* problem. These vendor lock-in problems can be resolved through the use of a multi-cloud storage system in which user’s data is not only bound to one storage provider but rather it is distributed over available multiple clouds. This also benefits the user in the reduction of user’s data storage and the increase in the availability of data over the cloud.

### 4.2. Bandwidth Utilization

The transmission of data over the network through some medium is known as bandwidth or sometimes referred to as throughput. Subjected to the other factors like latency, the performance could be better if the bandwidth capacity of the network is higher. Bandwidth demand has been increased in the last couple of years due to the reason that both the industrial and end-users have started relying more on the cloud storage instead of their in-house software and hardware. Due to which costs of bandwidth are increasing likewise other resources on the cloud [8]. Regardless of the heavy input/output (I/O) workloads and access latency, cloud providers have to maintain the sustainability level of I/O bandwidth of big data, which has now become a greater challenge for them. To achieve this challenge, cloud storage needs to run several processes in parallel which further require higher bandwidth utilization [9].

The total available bandwidth and the bandwidth utilized by the currently running application for a cloud user is the calculation criterion used by the Cloud Service Providers (CSPs) for customer bandwidth. Similarly, the bandwidth is optimized over the internet through online backups and several ways by CSPs. These ways used by CSPs keep on reducing the data been flowed through the internet bandwidth tunnels and techniques like deduplication and compressions over the cloud [10]. In this sense, reduction and minimization in the storage and bandwidth utilization prove to be more relevant as users are being charged according to their consumption [9].

### 4.3. Cost Effectiveness

The emergence of a newer cloud provider with a lower pricing scheme and more advance features remains an increasingly common occurrence in the cloud environment. This puts a cloud user in a decision-making state, either to transfer and migrate all of their data to the new cloud or remain with the existing one. Some may incur a high switching cost and migrate to other newer cloud while others may remain with the old ones and save the high cost of switching. Switching time from one cloud to another greatly affects the cost factor as switching earlier could be worse in some cases and switching later could have lost its value. Therefore, a user can benefit and take advantage of newer and old providers with best performances at a lower price through the distribution of their data over several cloud providers, which will entail only a fraction of data to be switched.

### 4.4. Storage Operations

All the pricing in the cloud storage is based on a flat rate of storage and a usage rate of the network. Therefore, the operations that are being performed or requested for the data stored over cloud storage play an important role in this regard. There are four main types of operation performed in cloud storage: POST, GET, PUT, and DELETE.

As it is already mentioned, the cloud storage providers choose their pricing according to the above-discussed operations; therefore, the selection of storage provider based on the pricing schemes of these operations play an important role.

### 4.5. Outages in Cloud Storage Availability

Most of the issues with the cloud storage data centers occur when they try to run heterogeneous multiple tasks and achieve high-level reliability at the same time [11]. The increase in frequency of these issues and failures causes outages in the cloud data centers. Most of the famous and large organizations like Facebook, Twitter, Dropbox, Google’s Gmail, Amazon, etc. are victims of these failures and outages [12,13].

According to a survey [14] in 2012, it was stated that due to the unavailability of services in 2007, there was a loss of US 1.7 million dollars; the average outage length per year was of 7.5 h, which resulted in the decrease of reliability as per demand or desired by the user. Major cloud providers have even gone through worse outages, because of which the users’ data have been corrupted or lost [15,16]. Incidents in the previous years show that some major cloud providers have also gone through several hours of outages, like PayPal in 2010 [17], Gmail with two outages in 2009 [18,19], and some others are listed below. The reason after the post-investigation of most of these outages revealed that the main root cause was the expected and predicted failures, while others happened due to the failure of correct components in the recovery process.

Similarly, most data loss occurs when the customers outsource their data to only a single cloud storage provider and lose control of their data. Therefore, whenever there is a failure to that single provider, then the user will not be able to access their data. Cloud of clouds is one of the solutions suggested in [20,21], which usually is a combination of diverse commercial clouds merged to form virtual cloud storage [22]. Most of the faults, outages, and failures continue to occur and are unavoidable even in presence of strict Service Level Agreements (SLAs) [23].

A report by the authors of [24] shows that even major cloud providers have gone through outages that resulted in bigger fail-over and some took much recovery time to get back to normal. In 2010, due to the positive response repetition in Skype, 30% of their supernodes stopped working and crashed, which made their overall system become overloaded. Whereas in the same year, 2010, one of Wikipedia’s datacenters was agitated and became offline due to the occurrence of a broken fail-over fault in the data center service. While in 2009, many of Gmail servers and data centers went offline during their maintenance and upgrading due to the bad request and cross data center routing. Twenty-five percent of the servers went down in the Google app engine service in 2010, due to power failure in their datacenter because of which all the user’s apps were in corrupted state. Due to network misconfiguration of nodes partitioning in the EBS cloud service, several of the clusters became distorted. In 2011, Paypal suffered from the global provision disruption due to its system’s network fault which happened because of the front end nodes becoming offline.

Keeping all of the above-mentioned outages issues and problems in mind, it is obvious that the user cannot be guaranteed long-term availability [22] for their data stored in cloud storage. Despite of the presence of very efficient and reliable availability techniques for cloud storage, the outages in cloud storage still happen and the reasons for these outages may vary from time to time. The following are the cases and causes of outages of some cloud storage providers which shows how even major storage providers suffered unavailability and had to bear the losses. 


**Outage Cases:**


On the 24 of February 2010, the Google app engine suffered from an outage that lasted for almost two hours [25]. According to the Google officials, the cause was the power failure in the primary Datacenter. The power failure along with the internal procedural issues rarely happens and because of which the recovery time of the Google app engine was extended.

Major online encyclopedia “Wikipedia” also suffered from a major outage on 24 of March 2010 which lasted many hours [26]. The reason for the outage was the overheating of one of its European datacenters due to which many servers turned off themselves to protect from the overheating effects. However, to keep the user’s traffic uninterrupted from this failure outage, it was forced to move all the traffic to the Florida datacenter. This diversion of whole traffic from the European datacenter to the Florida datacenter created the Domain Name System (DNS) change problem which also extended the recovery duration of outage failure.

One of the most used online peer-to-peer (p2p) communication service “Skype” also suffered from an outage of about 24 h from 22 to 23 of December 2010 [27]. The cause of this outage as officially reported was the overloading of support servers that support offline instant messaging. Due to this outage, enterprise users were not so affected unlike the end-users [28].

One of the Amazon web services (AWSs) known as Elastic Compute Cloud (EC2) also suffered major outages and failures twice in 2011 [29,30], which lasted not for hours but days. The reason for this major long-lasting outage was the non-execution of read and write operations [29]. In the very next year, 2012, Netflix, Instagram, Pinterest, etc. [31], which reside on EC2, also had unavailability due to the outage failure of Amazon EC2.

In February 2017 [32], the Amazon Simple Storage Service (S3) was disrupted and suffered from an outage which caused many websites and services to remain down for several hours. The main reason for this outage was the typo that was made incorrectly by the authorized member.

Therefore, achieving high availability is an ongoing problem in cloud storage [33]. Moreover, there are three basic major mechanisms used for the availability enhancement in cloud storage, i.e., replication, erasure-coding and deduplication, whose details are given in Section 5.

### 4.6. Multi-Cloud as a Solution to Single Cloud

Cloud-of-Clouds or Multi-Cloud are the terms that have been proposed by researchers to protect against the problems that were prompted in single cloud storage. These multi-cloud storage solutions provide so much agility to the user that they can assure their data is available all the time against the outages occurring in a single storage provider. The mobility of these multi-cloud storage systems is also observed in their redundant distribution over multi Geo-location through their redundancy mechanism. A multi-cloud solution also provides durability to data, fault tolerance, and avoidance from vendor lock-in problem through data stripping and distribution over multiple locations [23]. In such situations, comparatively erasure coding is found to be more significant as compared to replication because it minimizes stored data [34,35,36], but having read latency overhead.

According to the report conducted by 451 Research LLC on 1734 companies [37], one of the Microsoft Corporation commissioned studies, it was found that many new developments of hybrid cloud infrastructure are now emerging. Among this emergence, 33% of these host private clouds are in association with public CSPs, while 50% are public CSPs with on-premises private CSP, and 74% are hosted private CSP with on-premises private CSPs. A multi-cloud relates to cloud administration from numerous CSPs, while the hybrid cloud utilizes both private–public cloud administrations and capacities. Individuals will in general lean toward a multi-cloud system out of a wish to minimize their reliance on an individual CSP on account of vendor “lock-in” issues and the advantages of limited expense.

## 5. Mechanisms

### 5.1. Cloud Storage Mechanisms

It is understood from all the above discussion and cases that the outages that could last long are inevitable. Therefore, to make our cloud storage architecture tolerance free from the causes of outages, the following are the basic mechanisms used for the minimization of cloud outages and an increase in the availability of data over the cloud, whereas Figure 3 represents the hierarchy of Cloud Storage concerning availability mechanisms.

#### 5.1.1. Replication

Replication is a mechanism that works by creating multiple copies of original data and storing these copies or replications either within the same cloud, on geographical clouds, or both. Through the use of replication there is an increase in the availability of data, while, on the other hand, there is a reduction in the time latency of user data requests. Replication also reduces the bandwidth consumption made by the copies or replicas of the same data or service. This mechanism works by creating *N* copies of the data, where *N* > 1 always. These *N* copies or replicas make the data available all the time within the same cloud or on multiple geographic clouds. In the case where one of the clouds fails to provide the required replica due to any reason, then the replica from the other cloud is benefited to its user. This also ensures that the data is always in synchronization with the original data and with other geo-clouds. As it is unlikely that multiple clouds suffer from outages at the same time, the Inter-cloud replication is very effective in this manner [24,38]. The authors have proposed a 2-tier inter-cloud replication strategy through which they form a crash and outage-tolerant replication within the same cloud and then replicates across multiple clouds. To achieve this replication mechanism on a larger scale, the authors of [11,39] introduce a new replication algorithm based on Quality-of-Service (QoS) awareness. The proposed Quality-of-service-Aware Data Replication (QADR) [11] replication mechanism can handle and accommodate data-intensive applications on larger scales in cloud storage systems. The main problem that has remained open until now is the trade-off between the data available on cloud storage and the cost related to it. Therefore, the cost associated with the replica increments will also be increased which is not acceptable by the users. To maintain the data availability with the increase in replicas and reducing the cost caused by these replicas, the authors of [40] have proposed a low-cost replication scheme known as Multi-failure Resilient Replication (MRR). Their‘proposed mechanism handles both types of machines failures, i.e., correlated and non-correlated failures. Other than the mechanism that trade-off between replica increment and cost reduction, the authors of [33] have proposed a mechanism that utilizes the replication scheme along with load balancing architecture. Another mechanism which uses replication for the data availability along with the load balancing and cost reduction strategy is presented in [10].

**(A)*****Replication Types*** There are two main categories of replication schemes, i.e., *static replication* and *dynamic replication* [41]. Static replication well defines and predetermines the number of host nodes and replicas required for the availability [3]. In static replication, data is copied to a location, usually unchangeable, of other data centers. It also provides users with scalability across multiple locations because it only allows users to access data in an on-demand manner. Compared to static replication, dynamic replication provides an automatic creation and removal of replicas from the cloud storage based on the bandwidth, storage space, and the user’s access pattern [3]. Moreover, dynamic replication makes the decision based on the availability of resources. Due to the non-adaptive nature of static replication to the environment, it is not as much favorable and used as compared to dynamic replication. GFS [42], MinCopysets [43], RFH [44], MORM [45], etc. are some of the static replication scheme examples and RTRM [46], CDRM [47], CIR [48], D2RS [2], etc. are some of the dynamic replication scheme examples that are summarized in [3].

**(B)*****Replica/Data Selection*** In the replication scheme, the availability of cloud data depends on the basic three operations of a cloud storage system, i.e., which data or replica to be selected, where the replica/data is to be placed, and finally when the replica/data is to be placed. Initially, the replica selection plays an important role because the right selection of replica may reduce the waiting and response time of the requested data by the user while increasing the availability [49]. Moreover, the replica selection also helps out in the reduction of overall cost because the replica close enough to the user’s access pattern may be more selective as compared to the one which is far from the user. The authors of [50] have used Eucalyptus-based [51] availability in a cloud environment for their proposed techniques for replication selection. While the authors in [41] have proposed a replica selection technique based on the mechanism of ant colony optimization.

**(C)*****Replica/Data Placement*** As the number of data chunks or replicas in the replication mechanism is increasing for the achievement of high availability and reduced waiting time, the management of these replicas is also becoming critical accordingly. Moreover, increasing the number of replicas after some threshold will not necessarily increase the availability but will become the overhead. Therefore, in these scenarios, the location where the placement of replicas should be done becomes an important open issue that needs to be solved. However, the issue becomes more complicated when the replicas are to be placed in virtual data centers which are dynamically ultra-large and scalable [2,52].

Due to the socially-aware nature of the cloud, it becomes more challenging for the multi-clouds to achieve data/replica placement more conveniently. This type of socially-aware network over clouds requires that all related or friend’s data should be placed on clouds close to the user, so as to avoid the latency and availability issues. The authors of [53] achieve this type of data placement in a socially aware network through their proposed mechanism which uses data placement of multi-objectives. Along with the successful data placement, their proposed model also provides the cost minimization through graph cut technique.

The authors of [54] provide a new way of replica placement through the joint response time metric. Their proposed scheme works on peer-assisted cloud storage systems. While considering the complex requirements of multi-cloud storage systems, the authors of [23] have proposed Triones named model. For the formulation and optimization of data/replica placement in multi-cloud storage, this model uses nonlinear programming. Both types of requirements, i.e., simple and complex, are satisfied by the proposed model of Triones.

Many cloud storage providers use the Associated Data Placement (ADP) model proposed in [55]. The authors state that this ADP scheme tries to improve the location of associated data and the localized data so that to minimize the requested data latency and bandwidth utilization of the network while keeping the load-balanced over the cloud storage nodes. The authors of [56] have proposed a systematic model for the dynamic placement of data over distributed multi-clouds. Their model basically focuses on the management of big data over multi-cloud.

#### 5.1.2. Erasure Coding

Erasure coding is another availability improvement mechanism compared to replication used in the cloud storage systems. Despite the availability enhancements, it also reduces the cost of data storage over the cloud. The Basic working mechanism of erasure coding involves the breaking and dividing of data into *n* no. of blocks. The equation “*n = k + m*” satisfies this statement. Here, “*k*” is the original amount of data, whereas “*m*” is the redundant or extra data that is added to the original data for protection from failures. Here, “*n*” denotes the total data that needs to be stored over cloud storage after erasure coding process. Next, the encoded blocks are generated through “*n-k*” of “*k*” block, where “*n*” is the total number of blocks and “*n > m*”. All of these “*n*” blocks are then uploaded to exactly “*n*” no. of cloud storage where each cloud storage will be storing only one data block respectively. Making data available all the time through the elimination of corrupted data and reconstruction of original data by using the information stored elsewhere in the same cloud or in Geo-locations multi-cloud is the main goal of the Erasure Coding mechanism. Moreover, erasure coding also provides the reduction of total amount of data that is transferred over cloud storage. The general workflow of erasure coding is shown in Figure 4 and Figure 5.

There are many techniques [7,16,57,58] that have used erasure coding for the achievement of certain features along with the availability factor. However, on the other hand, most of these techniques lack in providing optimal solutions to other related issues. For instance, in Scalia [59], the erasure coding has been used for the optimization of cost factor. However, the work of the author is focused on single-objective optimization and lacks in multi-objective optimization, which makes it far away from the multi-cloud storage. Likewise, the techniques where erasure coding have been used for multi objective optimization [60,61] lacks in access latency, cost optimization, etc. factors. While the authors of [23] have also used erasure coding for the systematic data placement on multi-cloud storage systems. However, the drawback of erasure coding is that it can be more CPU-intensive, and that can translate into increased latency. While in [62,63], erasure coded methods are used for the cost effectiveness and high availability of intermediate data placement in multi-cloud storage. MTTDL is been used for estimating the lifetime of data in the multi-cloud. To optimize the problem of multi-objective, a Pareto-optimal set is used to minimize the intermediate data cost and maximize availability.

The authors of [64] propose an erasure coded fusion method which improves the recovery mechanism for the cloud storage systems. The EC-Fusion method overcomes the workloads problem of foreground and background processes and accelerates the recovery and response time while reducing the reconstruction time. Similarly, [65] also provides a recovery mechanism using pairwise balanced erasure coding design. The PBD-based erasure code method uniformly distributed the cross-rack traffic and retrieves a balanced number of available blocks for the recovery of bad or lost blocks.

#### 5.1.3. Data Deduplication

Data deduplication is a mechanism through which redundant data in cloud storage is eliminated and storage space is maximized. Basically, only the original data or one instance of redundant data remains on a single cloud storage media, whereas other multi-clouds only have the pointer referring to that original data or instance of redundant data.

The general workflow of data deduplication is depicted in Figure 6. In the first step of data deduplication, the data chunk or unit that needs to be checked and compared over cloud storage that whether it already exists or not is identified. After that in the second step hash value, also known as fingerprints, of these data chunks is created as their unique identification through some hash algorithms like Secure Hash Algorithm-1 (SHA-1), Message-Digest (MD5), etc. In the third step, duplicate data chunks are removed from the storage after careful checking of the hash values or fingerprints. In the fourth and last step of data deduplication, the unique data chunks whose fingerprints are found to be unmatched will be stored overcloud or storage devices and the index of these hashes will be updated accordingly.

Data deduplication has been categorized into two levels, i.e., file-level deduplication and block-level deduplication, in [66,67], respectively. Through the analysis of some security issues, the author of [66] has proposed a model that uses cross-user deduplication mechanism on the client side to minimize the risks of security and privacy. Moreover, two basic types of deduplication approaches, i.e., target-based deduplication and source-based deduplication, have been discussed in [66]. Similarly, scheme proposed in [68] also provide client side deduplication through the implementation of OpenStack Swift tool.

The models in [66,68,69] provide secure privacy preserving using deduplication scheme along with reduced storage on the client side. Similarly, the model in [70,71] also provides the security and confidentiality preserving deduplication mechanism for enhanced availability to the cloud data storage but with public auditing at enterprise level. The proposed model in [70] uses two level of deduplication: cross-user deduplication at enterprise-level and cross-enterprise deduplication which checks uploaded data by different enterprises. Basically, there are three major camps of deduplication, i.e., Storage, Data Distribution, and Network Communication [72,73]. The model in [74] uses deduplication mechanism for the validation of encrypted images stored on cloud storage. From all the state-of-the-art deduplication mechanisms it is identified that most of them work by identifying duplicates and eliminating the redundant data regardless of the level it is on through its collision-resilient fingerprints or more generally hash signatures that are cryptographically secure [75].

As mentioned above, for large volume of datasets, i.e., terabytes to petabytes, it becomes a bottleneck problem for the deduplication mechanism to store the hash values of each data chunk or block of such volume in memory. To overcome such a problem of storing hash values of huge data, the authors of [76] have proposed an AA-plus model. This model works in two ways: first the hash value of all the same applications is grouped together, and then after that, depending on the application type, the whole hash index is divided into groups. The MUSE model in [77] uses multi-tier SLA-driven deduplication along with the implementation dynamic deduplication regulation method to minimize the space cost and enhance the IO performance for large scale data. Similarly for the management of big data storage, the authors of [78] have proposed Distributed Deduplication with Fingerprint Index (DDFI) model which utilizes fingerprint index scheme with distributed deduplication. The DDFI model provided the use of minimum read/write bandwidth, minimum overhead over network bandwidth, etc.

Moreover, the distribution of total number of papers included in our literature review and analysis is depicted in Figure 7 in the form of Pie Chart Slices. Replication mechanism covers the lot of portion of this distribution which means that maximum methods and techniques have adopted replication mechanism for the data storage and availability achievement in the cloud. Whereas there are very few approached that use hybrid mode of storage mechanism in cloud. 75% of the research methods use at least one the major mechanism, i.e., replication, erasure coding, or deduplication, while others methods that do not fall in any of these covers 10–15% of the research methods included in our literature.

### 5.2. State-Of-The-Art Storage Mechanisms

#### 5.2.1. RACS

As switching between cloud providers in a multi-cloud environment is very expensive for the customers due to the lock-in problem, it becomes beneficial and success for some cloud providers. The authors of [7] present a proxy system that works by spreading the load of cloud users and customers over many cloud providers, thus eliminating the issue of cloud vendor lock-in. Their proposed model also provide cost efficiency and tolerance to the cloud outages as the model strips the data into multiple chunks and distribute over multiple clouds. The proposed model works similarly to the Redundant Array of Inexpensive Disks (RAID)-like structure through the use of disks and file systems but at cloud level, named RACS (Redundant Array of Cloud Storage). Moreover, the model works through implication of erasure coding mechanism to a different level of storage system in cloud platform.

#### 5.2.2. HAIL

Similar to the RACS in [7], the authors of [57] have also proposed a model that also utilizes the RAID related techniques for its employment. They named the model as HAIL (High Availability and Integrity Layer). The basic element that differentiates both Redundant Array of Independent Disks (RAID)-like structures is that RACS focuses on the minimization of economic failures and its prevention from excessive overheads. Along with providing high availability through the use of replication mechanism, the proposed HAIL model also provides security to the stored files and data over cloud storage. Some of the benefits of the proposed model include strong adversary, low overhead, static file protection, and strong file intactness. As the proposed model works only on static files, therefore it lacks in the operation of dynamic file environments.

#### 5.2.3. CDRM

The authors of [47] proposed a replication management scheme that dynamically and cost-effectively improves the performance and balances the load over cloud storage. Their proposed CDRM (Cost-effective Dynamic Replication Management) model works by maintaining the minimum replicas for the given availability requirement. CDRM uses Hadoop HDFS for its implementation. The placement of replicas depends on the node’s capacity and it’s blocking probability. The Workload on the nodes of cloud storage can be dynamically redistributable and adjustable to the cloud node’s capacity.

#### 5.2.4. CYRUS

The authors of [79] proposed a technique which use the concept of erasure coding to reliably store user’s data over multiple cloud providers. Their proposed technique CYRUS is a client-based mechanism that ensures that the privacy and reliability of the user’s data are maintained even after being stored over multiple clouds. Three types of criteria have been defined in the proposed model, i.e., privacy, reliability, and latency. The proposed model uses the basic APIs for accessing files, therefore it lacks in providing support to multiple clients if they update any file or information simultaneously. By splitting data and dispersing them over multi-cloud, CYRUS is ensured to be more consistent, reliable, and privacy-aware for the file and information storage over multiple clouds.

#### 5.2.5. μLibCloud

The authors of [16] have proposed a mechanism that is based on an Apache LibCloud library. Similar to the work in [79], the proposed model also makes use of erasure coding for splitting, dispersing, and collecting data to and from multi-clouds. The proposed model is a client-based library which works by encoding and decoding at the client-side and performing reading and writing operations at the cloud server end. Instead of reading from all the data chunks or stripes that are dispersed over multi-clouds, μLibCloud works only by reading from the most redundant and cheapest data sources. For the writing operation to be made successful, the client in μLibCloud has to write to all the repositories in the multi-cloud. Nine cities through VPS.NET [80] have been used as virtual machines in μLibCloud to make the access time for the client much lesser and minimized.

#### 5.2.6. HyRD

The authors of [81] proposed a model which uses the hybrid combination of replication and erasure coding mechanism to achieve higher availability and reduction in the redundant data. The data is dispersed over cloud provider’s storage after checking their workload diversity and characteristics. The proposed model HyRD (Hybrid Redundant Distribution) uses an erasure coding scheme for the distribution of large files in a cost-efficient way and uses replication scheme for small and metadata files for better performance of cloud storage providers. In this way, the model in [81] exploits the advantages of both replication and erasure-coded mechanisms and alleviates their disadvantages.

#### 5.2.7. DAC

The authors of [82] have proposed a deduplication model which provides better storage as well as availability solution for multi-clouds environment. Their Deduplication Assisted Cloud of Clouds (DAC) system makes use of deduplication mechanism which works by eliminating the redundant data from the distributed cloud environment. After the removal of redundant data, depending on the reference characteristics of the available multiple cloud providers, the original data is then dispersed among these providers. To make the model better for the availability, it also utilizes the benefits of replication and erasure code techniques. The highly utilized data is stored through replication while low referenced data is stored through erasure coding scheme in cloud storage.

#### 5.2.8. CHARM

As most of the time users store their data on single cloud storage, it puts them to a high risk of loss if that storage provider suffers from an outage or it raises the prices of its storage space, operations, etc. Keeping these types of issues in concern, the authors of [83] have proposed a novel approach through which they provide benefits in two phases. In the first phase, their model selects suitable cloud storage and some strategies through which it can store data at selected cloud storage with minimized cost and higher availability. While in the second phase the model provides its users with the beneficial functionality of triggering a transition process through which the data can be redistributed whenever the variation in the pricing of cloud or user’s access pattern is observed. Regardless of the cost efficiency and higher availability, the proposed model provides adaptability to the price and data variations.

#### 5.2.9. SCALIA

In [59], the authors have proposed Scalia, which is a client-based solution. Scalia works by observing the access pattern of the data and adapts the data placement accordingly. Three types of layers are there in Scalia, i.e., stateless engine, caching, and database. The most important engine layer of the system provides an API to all the cloud storage provider and also transparently works as a proxy between cloud storage and the client. Due to adaptive data placement accordingly to the stats of data access patterns, the proposed model cost-effectively satisfies the durability and availability constraints of cloud storage and makes the vendor lock-in problem avoidable.

#### 5.2.10. SPANStore

The authors of [36] have proposed a model with three key principles. The focus of the principles in the model is to minimize the cost factor and provide a fault-tolerant mechanism. The proposed SPANStore model first extents multiple cloud storage to increase the density of the cloud data centers that are distributed geographically. In the second principle phase, to satisfy the data propagation cost and the latency goals, SPANStore makes the trade-off between the geodistributed replications through their application workload estimation. In the last principle phase of SPANStore, the implementation of two-phase locking and data propagation task is done through the minimization of computing resources. For the cost minimization, SPANStore also helps in determining where to replicate and how to replicate. SPANStore states four goals to achieve, i.e., cost and latency minimization, flexible consistency, and fault tolerance [36].

#### 5.2.11. Syncopy

Syncopy [84] lets the user decide which files are to be replicated over clusters. Authors have to make a little modification in the code of the distcp tool of HDFS so as to retrieve the destination location of cluster. Syncopy only updates or transfers the newly added data to remote clusters. While mirror replication model by [85] provides proof of retrievability which advantages by resisting the network attack, high security achievement, efficient verification, etc. to enhance cloud storage availability.

#### 5.2.12. Rejuvenation Mechanism

The authors of [86] have proposed a model in which a live migration of VMs to other hosts is done through a rejuvenation mechanism. This rejuvenation process for the live migration of VM is done in a time-based vector. Authors have used the rejuvenation as a fault preventing technique to achieve higher availability. There are four components of the proposed model: management server, main node, standby node, and the remote storage. The advantage of the proposed rejuvenation model is that it does not go down due to the aging factor or due to any hardware/software faults, because if anything happens to the main node then VM will be migrated to the standby node and it will work as the main node then. The proposed model [86] is also beneficial in achieving high availability for the cloud storage systems where aging is faster due to heavy workloads.

#### 5.2.13. IDO

There are two types of tasks when talking about storage systems, i.e., high priority tasks and low priority tasks. High priority tasks are run in the foreground, while low priority tasks are run in the background. The authors of [87] have proposed an Intelligent Data Outsourcing (IDO) optimization technique whose focus is on improvement and enhancement of low priority tasks. This technique works proactively by migrating the data of hot data zones to a surrogate RAID structure from a degraded RAID structure before the occurring of any failure event, while other techniques work reactively which also slows down the addressing of the I/O request from the degraded storage. IDO improves the execution of these tasks so to provide higher data availability on the user’s I/O request even if the storage system is degraded.

#### 5.2.14. PRCR

The authors of [88] have proposed the PRCR model, which cost-efficiently stores minimum replicas of data while ensuring the reliability of it. The PRCR model allows reliability management along with the minimum replica storage and negligible operational cost. Reliability management provided by PRCR is with a variable disk failure rate while explicitly presenting the benchmark of minimum replication. PRCR is served by the cloud storage providers as a service that runs and operates on Cloud’s virtual machines.

### 5.3. Cloud Storage w.r.t Healthcare Systems

It is promising for the medical care organizations to consider receiving healthcare data innovation frameworks because of the ascent in the expenses of medical care administrations, medical services experts are becoming limited and difficult. To provide more effective and efficient medical and health services, HIT allows healthcare institutes and organizations for streamlining many of their processes. For making true enabling of HIT services over the internet and cloud, it makes use of latest technological trends like Cloud Storage Computing (CSC) infrastructure. In the following subsections we have discussed and highlighted some of the constituents and proposing the building of cloud storage over the e-Healthcare environment.

#### 5.3.1. Healthcare with Big Data

There is a huge volume of information in social insurance that is identified with various medicinal services areas particularly neuro and heart. This information needs an extraordinary concentration and the models right now concentrating on these areas needs to actualize the most recent advances to foresee a few examples. The authors of [89,90,91,92,93] emphasize various social insurance framework designs that are utilized to accumulate live information from everywhere throughout the world. For this, they have utilized AI components and approaches of Big Data to propose cross breed information forecast and taking care of procedure. While authors in [94,95,96,97] have over viewed and summed up how medicinal services frameworks are carefully changing using clinical innovation, wearable brilliant gadgets, computerized clinical records, and so on. The job of clinical huge information turns into a difficult assignment as capacity, required data recovery inside a constrained time, cost proficient arrangements in wording care, and numerous others. The examinations by these creators help in the ID clinical huge information includes the use of clinical large information, and the investigation of the huge information in cardiology. Their examination is useful for the specialists, experts, clinical doctors, and so on for settling on more true prescription and choice for the fix of illnesses.

#### 5.3.2. Healthcare with IoT

Blending the Internet of Things (IoT) with building data displaying can improve the presentation of the information assortment. The authors of [98] have utilized an indoor situating framework in their model for the assortment of information from certain medical clinics to decide the success of IoT gadgets introduced in the premises of the emergency clinic. Their information is useful for the social insurance association to have an all encompassing perspective on clinic premises on 2D map. While the authors of [99] utilize wearable savvy IoT gadgets for the wellbeing checking of the patients. These IoT gadgets are straightforwardly associated with the electronic medicinal services frameworks which encourage both the patient and the specialists for observing on ordinary premise without having any physical cooperation. Different exploration points of view identified with security and protection inside IoT-cloud-based e-Health frameworks are inspected, with an accentuation on the chances, advantages, and difficulties of the usage of such frameworks [100,101]. The utilization of Artificial Intelligence (AI) has changed the human services based IoT frameworks at pretty much every level. The mist/edge worldview is bringing the registering power near the conveyed arrange and thus alleviating numerous difficulties in the process [102]. The blend of IoT-based e-Health frameworks incorporated with canny frameworks, for example, distributed computing that give savvy targets and applications is a promising future pattern.

#### 5.3.3. Healthcare with Machine Learning

The work in [103] distinguishes the patterns and ways to deal with artificial intelligence research in medicinal services. Their community-oriented work significantly improves the detailing of hypothetically pertinent systems to control observational examination and application, especially important in the quest for causal components to decrease exorbitant and avoidable medical clinic re-admissions for constant conditions. While it has stayed a major test for the exploration network to build up a finding framework to recognize diabetes effectively in the e-Healthcare systems. The current determination frameworks utilize high calculation time and low forecast exactness. In this way, the authors of [104] proposes a determination framework utilizing AI strategies for the recognition of diabetes. They have utilized Ada Boost and Random Forest calculations for the component’s choice. The proposed strategy exhibits that it could adequately distinguish diabetes and can be conveyed in an e-Healthcare services condition. Though, the creators in [105] propose a productive and precise framework to determine coronary illness and the framework depends on AI procedures. The proposed arrangement of [105] is created dependent on the characterization calculations of AI. Highlights are chosen based on restrictive common data, which builds the arrangement precision and limits the handling season of framework.

### 5.4. Critical Evaluation

There are many techniques available for the enhancement and optimization of cloud storage availability, but Table 2 and Table 3 illustrate some of these different state-of-art mechanisms of cloud storage availability related to our focused study. The most influential techniques have focused on the storage space and cost minimization along with providing high availability through the use of data deduplication and erasure coding respectively. DAC [82], RACS [7], and Scalia [59] are most prominent among this category, whereas replication is also used by most of the techniques by making trade-off between availability, storage space, and cost minimization. GFS [42], CDRM [47], and PRCR [88] are some common techniques of this category. Moreover, some techniques use hybrid approaches through minimizing the drawbacks of some and taking advantages of other, like DAC [82] and HyRD [81].

The overall purpose of all the techniques mentioned in Table 3 is to analyze them based on of maximization in availability of data in cloud storage and minimization in outages, vendor lock-in, extensive storage cost, etc, whereas Table 2 represents and evaluates the techniques based on year, approach used or adopted, and their operating type. The techniques in Table 3 have been compared and evaluated based on multiple characteristics: Availability, Reliability, Response Time (RT), Cost Minimization (CM), and Recovery. Availability, Reliability, Cost Minimization, and Recovery characteristics are measured in scale of High, Moderate, and Low, whereas Response Time is measured in the scale of Maximum, Moderate, and Minimum.

## 6. Limitations of Our Work

Following are some of the limitations to the validity of our proposed work.

The work presented in this paper through the SSLR process is limited to only six most extensively used repositories, whereas there exist number of digital repositories for accumulating the research articles.There are continuous publications in the related domain on daily basis, but our current work only follows the research done in years ranging from 2007 to 2020.There may be articles, that have been skipped, while they contained the keywords healthcare and cloud storage but for the implementation purposes they have no concern.Fundamental purpose of skipping Google Scholar in our research is to save time from getting difficulty in the duplicate entries.A large number of research has been published in the related domain but the authors limited their research through their defined set of keywords. More keywords can be defined to enlarge the work in the future.

## 7. Limitations of Healthcare Cloud Storage Domain

### 7.1. Implementations

There are basic problems that each medical consideration affiliation must suffer when adapting to cloud-based services. Turning over data, security, availability, and control to an outcast suggest that your association has unquestionably no impact over where its data truly lives. Trust in your cloud dealer takes on a whole new significance. Security and insurance, which are key issues in the clinical consideration industry, must be invulnerable.

Changing from an on-premises foundation to the cloud suggests changing your entire methodology for dealing with tasks. Clinical administrations providers needing to realize a cloud game plan must ensure that everybody comes up to speed with how to go after the cloud capably. Something different, your business chances individual time, rash treatment of data, or information spills.

### 7.2. Availability and Control

Despite all odds, distributed storage will go down occasionally. Clinical consideration providers need their data to be available whenever, so any get-away on the distributed storage side will adversely influence productivity. This is substantial for business-had physical foundations likewise, anyway associations must rely upon the cloud provider—not themselves—to bring the organization back on the web.

### 7.3. Compliance

All distributed storage based medical services associations must consent to the Health Insurance Portability and Accountability Act (HIPAA). This consolidates healthcare endeavors, yet also loosens up to shows for open minded security, approval of laws, and break cautioning technique. The inhabitants of HIPAA ought to be seen by both the clinical consideration and cloud suppliers in order to ensure HIPAA consistence.

### 7.4. Security

Distributed storage networks give security devices that expect to, alert you of, and oversee obscure cloud services. In any case, they are not incredible. The U.S. Part of Health and Human Services’ Office for Civil Rights is presently exploring cases including security breaks of cloud storage based-medical care information. Of those cases, 47% were achieved by hacking or an IT event. Most vendors will have unquestionably a bigger number of limits than the individual customer. Unapproved presentation of information achieves genuine results to the affiliation and significant costs in recovering and restoring data similarly as exhorting impacted individuals.

## 8. Research Challenges

### 8.1. Security-Privacy Concerns

The cloud storage-based healthcare is incredibly defenseless against assaults for a few reasons. To begin with, frequently its segments invest the greater part of the energy unattended; and consequently, it is anything but difficult to assault them truly. Second, the greater part of the interchanges is remote, which makes listening in amazingly basic. At long last, the majority of the cloud storage-based healthcare segments are portrayed by low capacities with respect to both vitality and processing assets and hence, they cannot execute complex plans supporting security. All the more explicitly, the serious issues in cloud storage-based healthcare are identified with security concern confirmation, and information respectability. Verification is troublesome as it for the most part requires fitting validation frameworks and workers that accomplish their objective through the trading of suitable messages between different hubs. Common cryptography calculations spend a great deal of assets with respect to vitality and data transfer capacity, both at the source and the objective. Such arrangements cannot be applied to the cloud storage-based healthcare, given that they will incorporate components, which are genuinely compelled with respect to vitality, interchanges, and calculation abilities. It follows that new arrangements are required ready to give a palatable degree of security [109,110].

Clinical and medical administrations data not in any way like various kinds of data have serious mystery, assurance, and security concerns. HIPAA consistency is the most significant essential while moving clinical records to the cloud. Moving entire data amassing to a outcast affiliation is certifiably not a basic endeavor to do, especially while moving sensitive information, for instance, clinical consideration data. Altogether more solid security should be ensured in light of the fact that more concerns will develop with access controls, survey controls, affirmation, endorsement, transmission security, and limit security in order to swear off introducing the information to unapproved components. These issues are an obstacle that has moved back cloud determination and should be tended to in order to enable the trustworthiness of cloud structures. Fortunately, immense quantities of the best cloud providers in the market, for instance, Microsoft, Google, and Amazon have obligations to develop the best methodologies and practices to ensure about a customer’s data and assurance [111,112].

### 8.2. Heterogeneity

A major search in the cloud-based healthcare is identified with the wide heterogeneity of devices, working frameworks, stages, and administrations accessible and potentially utilized for new or improved applications. Cloud storage platforms heterogeneity is likewise a non-irrelevant concern. Cloud benefits regularly accompany exclusive interfaces, causing asset mix and solution to be appropriately redone dependent on explicit suppliers. This issue can be exacerbated when clients receive multi-cloud draws near, i.e., when administrations rely upon different suppliers to improve application execution and flexibility. These angles are just halfway tackled by cloud expediting, deliberately executed by cloud services or by outsourcing organizations.

### 8.3. Interoperability and Standardization

Interoperability is probably the greatest test while moving medical care frameworks to the cloud. It is because of the huge presence of various conventions, operating systems, programming dialects, stages, information configurations, data sets, and approaches that distinctive medical care associations have been utilizing. Medical services frameworks are not at present planned utilizing normal information demonstrating develops bringing about various information base plans [113] and in congruent frameworks.

Medical care frameworks interoperability must happen in a few unique manners: at the supplier, programming, information levels, and framework reconciliation. Suppliers have commonly kept up their own free information and the contrariness of medical care frameworks generally forbids its cross-institutional use [113]. To grasp the cloud storage-based healthcare, associations must coordinate their current frameworks with present day web and cloud-based frameworks. Also, they ought to normalize cycles, for example, the way toward getting patient’s data and sparing it to the distributed storage.

Another way to deal with creating medical services frameworks ought to be taken so as to plan more interoperable frameworks. This change will bring about various and significant advantages to the healthcare network. Incorporating current medical services frameworks and making them interoperable with the freshest cloud-based advancement seems to be a difficult assignment. Albeit, planning adaptable and versatile guidelines and incorporating clinical information will enormously profit and help the different parental figures.

### 8.4. Standardization

As the quantity of cloud based devices soared, the standardization concern has emerged. The standardization issues are referenced when cloud based devices are applied to a wide scope of orders that are constrained by various administrative gatherings. On account of cloud based storage and distributed computing in medical services, the intricacy turns out to be all the more testing because of the rigid guidelines and clinical norms. Thus, it is vital that cloud based devices producers and diverse administrative gatherings need to set up standard strategies and rules to ensure standardization.

### 8.5. Monitoring

Research is as yet required on the execution and advancement of appropriate correspondence conventions, the setting of different cloud-based healthcare standards for advancing interoperability and for scaling the expense of medical services offices, and the evaluation of dangers and vulnerabilities. Additionally, cloud-based medical care acquires similar checking prerequisites from Cloud; however, the related difficulties are additionally influenced by volume, assortment, and speed attributes of medical services.

### 8.6. Social and Legal Aspects

There are two significant challenges that are incompletely related. Legal perspectives are critical and genuine in the momentum research for explicit application situations. The expert co-op needs to adjust to various global laws. Additionally, clients must be furnished with motivators to add to information assortment. Clients could likewise be engaged with new structure squares and devices: quickening agents, systems, and toolboxes that empower the investment of clients in cloud storage based-healthcare as done on the Internet through Wikis and Blogs [114,115,116,117]. Such devices and methods should empower analysts and plan experts to find out about client work, giving clients a functioning part in innovation plan.

### 8.7. Transition Process

Medical care organizations coordinate cloud-based storage and distributed computing in existing medical services frameworks by supplanting or including clinical gadgets and sensors into the current gadget organization. Nonetheless, gadgets from various merchants have completely extraordinary correspondence conventions. Thus, it is a test to guarantee a smooth change of these new gadgets. Accordingly, it is obligatory that producers and merchants observe similar norm to ensure that their gadgets uphold in reverse similarity when they are conveyed on a current organization of gadgets.

### 8.8. Big Data Network and Analysis

With an expected number of 50 billion gadgets that will be organized by 2021 [118], explicit consideration must be paid to transportation, storage, access, and handling of the tremendous measure of information they will deliver. On account of the ongoing improvement of advancements, cloud storage will be one of the principle healthcare of large information, and cloud will empower to store it for quite a while and to perform complex examinations on it. Taking care of the information is a basic test, as the general application execution is exceptionally reliant on the properties of the information the board administration [119,120].

The unpredictable idea of information gathered from wearable gadgets and sensors is another trouble. The intricacy increments when the pace of information created is rising. The executed framework must get ready for the information unpredictability challenge by zeroing in additional on haze registering layers to build the processing force, and utilizing the assets with effective information preprocessing and information examination calculations.

## 9. Future Open Directions

Numerous works from scientists, merchants, and governments have been dedicated to making and creating novel cloud storage and healthcare applications. Alongside the momentum research efforts, we empower more bits of knowledge into the issues of promising coordinated innovation, and more directions intending to the open research problems in this paper. In the following, we feature some future research directions in the domain of Cloud storage based-Health applications. An assortment of innovations that are required in Cloud storage based-Health frameworks are as per the following.

### 9.1. Data Warehouse

The gathered datasets by distributed storage registering ought to be put away and documented shaky, simple to utilize and dependable information bases and information stockroom to guarantee that the trustworthiness and security of the information and datasets can be kept at the ideal. As not all the information and datasets can be utilized each time, a smart framework is required in the information distribution center to pick the correct information and datasets for each assistance. Large information is a significant exploration point that is firmly combined with Cloud storage based-Health and with a few related difficulties.

### 9.2. Machine Learning Techniques

Computerized reasoning and AI give the premise to large information handling since enormous information administrations require clever calculations [121], frameworks, and administrations to deal with a large number of informational collections, comprehend the connection between every single distinctive variable, measure all the necessities and present all yields [122]. The test for the following stage is to plan prescient frameworks for early discovery of certain maladies and illnesses.

### 9.3. Standardization

The absence of guidelines is considered as a major issue towards Cloud storage based-Health mix by countless scientists. As of now, most things are associated with the Cloud through electronic interfaces, which can lessen the unpredictability of growing such applications. Notwithstanding, they are not explicitly intended for effective machine to machine correspondences and present overhead with respect to arrange burden, postponement, and information preparing. In addition, interoperability is as yet an issue, on the grounds that both the cloud and the things actualize non-standard heterogeneous interfaces.

### 9.4. Storage

Cloud storage plans have concisely been considered in this paper. For example, we have recently considered them as an operator for the coordination of cloud storage based-Healthcare. Regardless, the literature considers this as a notwithstanding all that open issue as current plans may not offer significant assistance for future directions. One possible bearing to address such issues incorporates the introduction of perceptive storage.

### 9.5. System and Software Architecture

Framework and programming structures should be characterized, coded, and executed dependent on the standard, for example, lithe turn of events, Map-Reduce system, and any product structures that can diminish the preparing time and intricacy. Some information investigation despite everything relies upon factual strategies to decipher the implications of relapse and displaying yields. Factual investigation can perform relapse tests and present key yields [123].

### 9.6. Power and Energy Efficiency

Ongoing Cloud storage based-Health applications incorporate progressive data transmission from the things to the cloud, which, accordingly, may rely upon mobile phones as an entry. Such a cycle quickly drains battery limit on both the things and the entry confining the predictable movement to 24 h or less. The writing shows that, in the field carefully identified with the joining of Cloud storage and medical services, acquiring vitality proficiency in both information handling and transmission is a significant open issue.

## 10. Conclusions

The e-Health Cloud shows an enabling development for some health and medical consideration providers to face various troubles, for instance, rising clinical consideration movement costs, information sharing, and insufficiency of clinical administration specialists. Regardless, the points of interest got are adjusted by issues of trust, assurance, and security aside a couple of particular issues that must be tended to before clinical administrations providers can get trust in the e-Health Cloud. A detailed literature review of cutting-edge cloud storage and medical care components and their related issues was introduced in this paper with an accentuation on the significance of the ideas in concepts, usage, and difficulties. There has been a lot of interesting work done for achieving the higher availability of cloud storage in healthcare systems, which have been compared and critically analyzed in this paper. Among all the currently present state-of-art mechanisms, there is not a single mechanism that could have solved the all availability issue along with the cost, performance, reliability, and storage space efficiency factors. This study also helps to identify the future research challenges for the development of such methods and framework models that could resolve the reliability and availability issues of the current mechanisms in the healthcare environment.

The current study will assist the specialists with a helpful base for future work to comprehend the general setting of medical and health services and its applications in distributed cloud storage. In the future, we will likewise attempt to build up a definite report of cutting edge procedures utilized in the investigation of clinical and health care distributed cloud storage information generally in medical services and explicitly in neurology science. This will push the experts to effortlessly uphold dynamic in medical services.

## Figures and Tables

**Figure 1 sensors-20-05392-f001:**
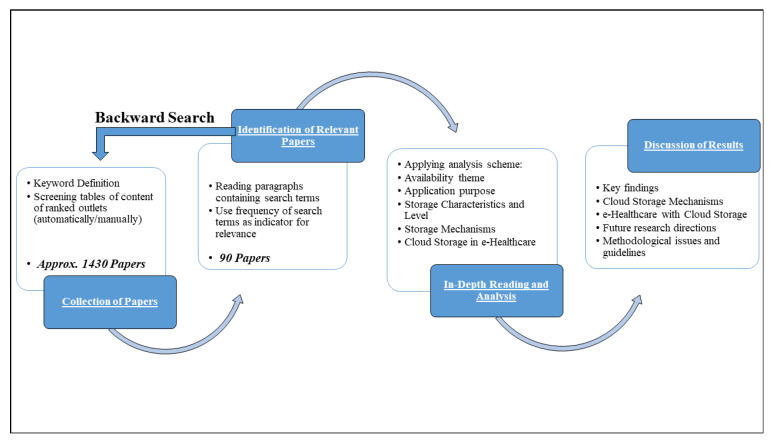
Strategic workflow of the paper’s literature and analysis.

**Figure 2 sensors-20-05392-f002:**
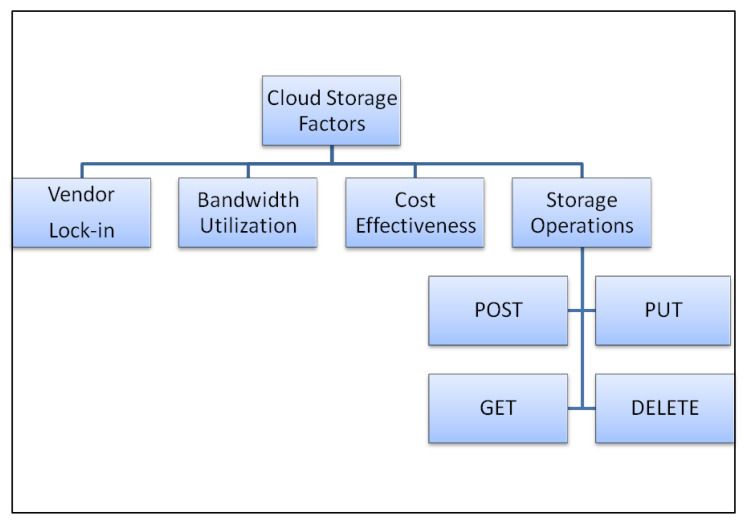
Basic factors of cloud storage.

**Figure 3 sensors-20-05392-f003:**
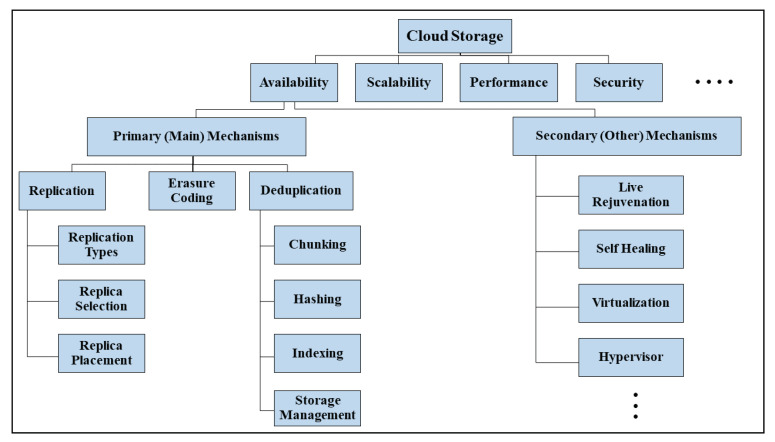
Hierarchy of cloud storage availability and its mechanisms.

**Figure 4 sensors-20-05392-f004:**
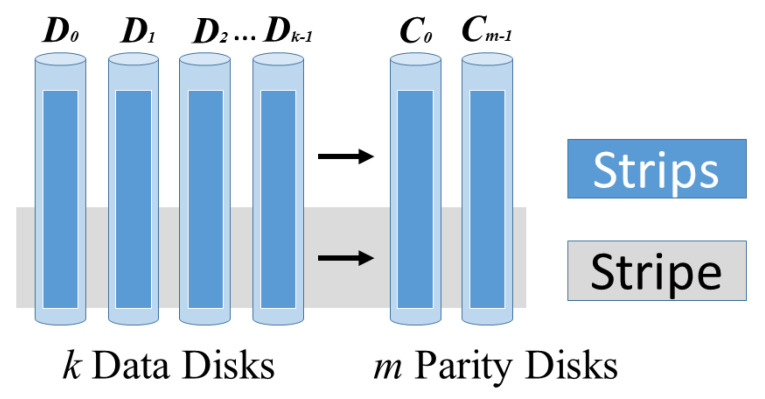
Erasure coding general workflow.

**Figure 5 sensors-20-05392-f005:**
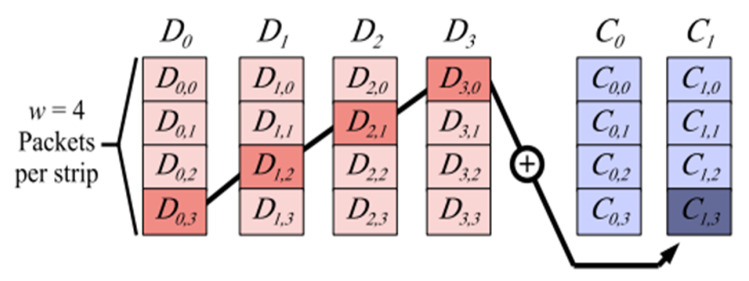
Erasure coding working example with *k = 4*, *m = 2*, and *w = 4*.

**Figure 6 sensors-20-05392-f006:**
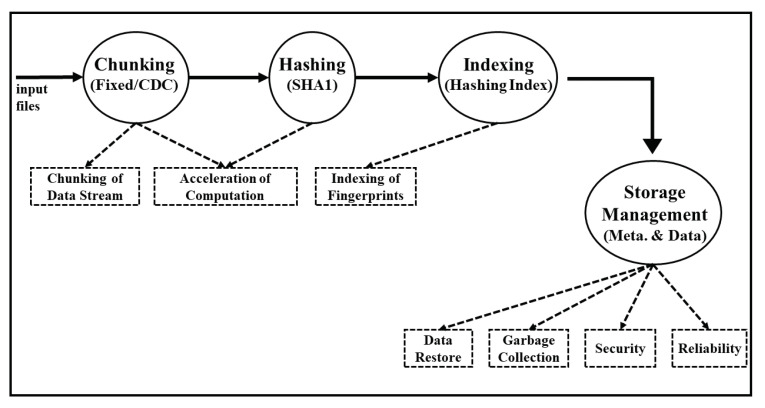
Workflow of data deduplication.

**Figure 7 sensors-20-05392-f007:**
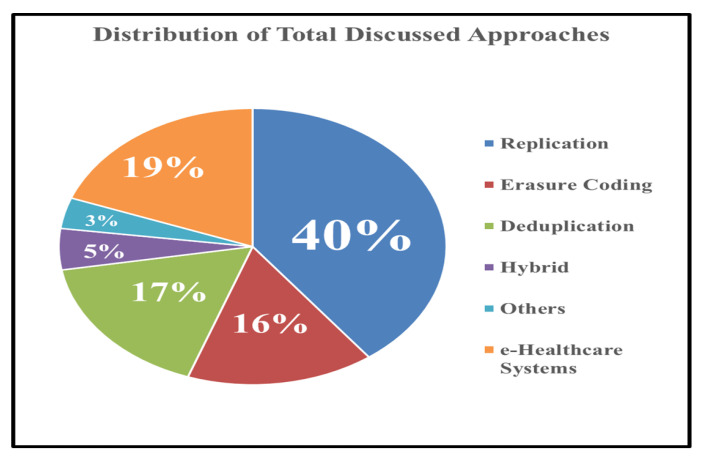
Pie chart representation of distribution of total discussed approaches based on primary and secondary mechanisms.

**Table 1 sensors-20-05392-t001:** Distribution of total number articles included in this paper’s literature based on the Year and their concept methodology.

Year	Methodology								
	*Action Research*	*Archival Research*	*Conceptual*	*Case Study Research*	*Field Experiment*	*Field Study*	*Ground Theory*	*Lab Experiment*	Total
2007 & Before	-	-	3	2	1	-	-	-	**6**
2008	-	-	-	1	-	-	-	-	**1**
2009	-	-	-	-	-	-	-	1	**1**
2010	1	-	1	3	2	-	-	-	**7**
2011	-	-	-	2	-	1	-	-	**3**
2012	1	-	2	1	-	1	-	1	**6**
2013	2	2	-	1	-	2	3	1	**11**
2014	-	-	1	2	-	3	2	-	**8**
2015	-	-	2	2	-	-	-	2	**6**
2016	2	-	2	3	-	2	-	1	**10**
2017	-	-	1	2	-	-	-	-	**3**
2018	2	1	-	2	-	2	1	-	**8**
2019	1	1	-	1	-	1	1	-	**5**
2020	4	5	3	3	3	2	1	1	**22**
**Total**	**13**	**9**	**15**	**25**	**6**	**14**	**8**	**7**	**97**

**Table 2 sensors-20-05392-t002:** Evaluation table of the techniques and methods on the basis of their operation type and approaches used.

Methods	Reference	Approach	Operating Type
DAC	[82]	Replication; Erasure Coding (*for Availability*); Deduplication (*for Storage*)	Client-Based
HAIL	[57]	Replication	Cloud-Based
GFS	[42]	Replication	Cloud-Based
HDFS	[84]	Replication	Cloud-Based
Syncopy	[106]	Replication	Cloud-Based
CDRM	[47]	Replication	Cloud-Based
RACS	[7]	Erasure Coding	Client-Based
*μ*LibCloud	[16]	Erasure Coding	Client-Based
HyRD	[81]	Replication (*for Small Files*); Erasure Coding (*for Large Files*)	Client-Based
PRCR	[88]	Replication	Cloud-Based
CHARM	[83]	Replication; Erasure Coding	Cloud-Based
Scalia	[59]	Erasure Coding	Client & Cloud-Based
AA-Plus	[107]	Deduplication	Cloud-Based
Kaaniche et al.	[68]	Deduplication	Client-Based
CYRUS	[79]	Erasure Coding; Deduplication	Client-Based
IDO	[87]	Other (*Proactive Data Migration*)	Cloud-Based
Self-Healing	[108]	Other (*Self Healing*)	Cloud-Based
Live Rejuvenation	[86]	Other (*Live Rejuvenation*)	Cloud-Based
J. Liu et al.	[40]	Replication	Cloud-Based
SPANStore	[36]	Replication	Cloud-Based

**Table 3 sensors-20-05392-t003:** Critical comparison and evaluation of storage mechanisms on the basis of their approach used and some focused characteristics.

Methods/Reference	Focused Characteristics with Importance Measure				
	*Availability*	*Reliability*	*Response Time*	*Cost Minimization*	*Recovery*
DAC-[82]	High	Moderate	Minimum	High	High
HAIL-[57]	High	Low	Moderate	High	Moderate
GFS-[42]	High	High	Minimum	Low	High
HDFS-[84]	High	High	Maximum	Low	Moderate
Syncopy-[106]	High	High	Minimum	High	Low
CDRM-[47]	High	Low	Minimum	Moderate	Low
RACS-[7]	High	High	Maximum	Moderate	High
*μ*LibCloud-[16]	High	Low	Moderate	Low	Moderate
HyRD-[81]	High	Low	Moderate	High	High
PRCR-[88]	Moderate	High	Minimum	High	High
CHARM-[83]	High	Low	Moderate	High	Low
Scalia-[59]	High	Moderate	Minimum	High	High
AA-Plus-[107]	Low	High	Moderate	Moderate	Low
[68]	Moderate	low	Minimum	Low	Moderate
CYRUS-[79]	Low	High	Minimum	High	Moderate
IDO-[87]	High	High	Minimum	High	High
Self-Healing-[108]	High	Moderate	Moderate	Low	High
Live Rejuvenation-[86]	High	Low	Moderate	Low	High
[40]	High	Moderate	Moderate	High	Moderate
[36]	High	Moderate	Minimum	High	Low

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
