# Peer review of "A Systematic Review on Cloud Storage Mechanisms Concerning e-Healthcare Systems"

_sensors, 2020, doi:10.3390/s20185392_

Round 1

Reviewer 1 Report

The authors have provided a systematic review of cloud storage services. The authors have started with keyword search and it was followed by screening table of content of ranked outlets. The authors have sorted the papers based on the frequency of the search terms in the papers. At this stage, they have reduced the number of papers to 90 from 1430 papers.  They have focused on the availability of the cloud storage services. The paper discussed about different storage characterises and mechanisms to improve the availability of the storage service. Finally, they have highlighted the importance of cloud storage in e-healthcare. The paper concludes with the research direction. However, the paper suffers from several limitations, such as:

  1. Though the paper promises to discuss different aspects of cloud storage mechanism with respect to e-healthcare, but the authors have failed to provide interesting insights. They have only discussed few points about healthcare in page 22 and 23. From the discussion, it is difficult to follow what are the specific requirements of e-health sector in terms of Cloud storage.
  2. The discussion in section 4.2 about e-healthcare systems looks misplaced. The authors need to re-write the paper if they want to include the e-healthcare focus. Otherwise, they can just focus on the availability aspect of cloud storage service.
  3. It is not clear from the paper, which academic databases are used to search the papers. The authors have also failed to discuss the questions which provide the guidelines for any systematic literature survey.
  4. Limitations of the existing solutions and the future research direction should be more emphasized.
  5. There are several review papers available on theis topic, the authors can review and compare those papers with their paper and highlight their contribution.
  6. Section 3 can be removed or merged with another section.
  7. The paper needs a major proof reading.

Reviewer 2 Report

  1. The manuscript presents a systematic review on enabling mechanisms of cloud storage availability for e-Healthcare systems, which is interesting. The subject addressed is within the scope of the journal.
  2. However, the manuscript, in its present form, contains several weaknesses. Appropriate revisions to the following points should be undertaken in order to justify recommendation for publication.
  3. Full names should be shown for all abbreviations in their first occurrence in texts. For example, NIST in p.1, IT in p.1, CSPs in p.4, QADR in p.9, etc.
  4. For readers to quickly catch your contribution, it would be better to highlight major difficulties and challenges, and your original achievements to overcome them, in a clearer way in abstract and introduction.
  5. It is shown in the reference list that the authors have several publications in this field. This raises some concerns regarding the potential overlap with their previous works. The authors should explicitly state the novel contribution of this work, the similarities and the differences of this work with their previous publications.
  6. It is mentioned in p.1 that e-Healthcare system is adopted as the focus of this study. What are the other feasible alternatives? What are the advantages of adopting this field over others in this case? How will this affect the results? More details should be furnished.
  7. It is mentioned in p.3 that four keywords are adopted to carry out automatic in-text search engine screening or manual screening. What are other feasible alternatives? What are the advantages of adopting these keywords over others in this case? How will this affect the results? The authors should provide more details on this.
  8. It is mentioned in p.7 that a scientometric analysis, similar to the one as represented by Sabine et al in [35], is adopted in this study. What are the other feasible alternatives? What are the advantages of adopting this approach over others in this case? How will this affect the results? More details should be furnished.
  9. It is mentioned in p.18 that some mechanisms of cloud storage availability as shown in Table 4 and Table 3 are adopted for critical evaluation. What are other feasible alternatives? What are the advantages of adopting these mechanisms over others in this case? How will this affect the results? The authors should provide more details on this.
  10. This is a review paper regarding enabling mechanisms of cloud storage availability for e-Healthcare systems. What are the novelties that can be brought out by reviewing this specific topic?
  11. What are the major differences between this review and other earlier review papers on this topic?
  12. More critical analysis should be made to different modelling methods, with both advantages and disadvantages of each method explicitly exhibited.
  13. A more balanced distribution on the description amongst different machine learning models should be made.
  14. More details should be furnished on different hybrid models. More critical comparison should be made on this issue.
  15. Some assumptions are stated in various sections. Justifications should be provided on these assumptions. Evaluation on how they will affect the results should be made.
  16. The discussion section in the present form is relatively weak and should be strengthened with more details and justifications.
  • Moreover, the manuscript could be substantially improved by relying and citing more on recent literatures about real-life applications of machine learning techniques in different fields such as the followings: https://doi.org/10.1080/19942060.2018.1452296
  • https://doi.org/10.1080/19942060.2017.1330709
  • https://doi.org/10.1080/19942060.2017.1335653
  • https://doi.org/10.1080/19942060.2018.1474806
  1. Some inconsistencies and minor errors that needed attention are:
  • Replace “…minimizing the its outages…” with “…minimizing its outages…” in line 8 of p.1
  • Replace “…than 3 times in there literature content…” with “…than 3 times in their literature content…” in line 68 of p.3
  • Replace “…that single provider, than the user will not be able…” with “…that single provider, then the user will not be able…” in line 153 of p.5
  • Replace “…in a cost efficiently way…” with “…in a cost-efficient way…” in lines 616-617 of p.19
  1. In the conclusion section, the limitations of this study, suggested improvements of this work and future directions should be highlighted.

Round 2

Reviewer 1 Report

Thank you for accommodating the feedback. I am happy with the changes. However, I would like to suggest that a more in-depth discuss about the limitations of the current research proposals (not the limitations of your study) and future research direction in cloud storage for e-health application will help other readers.

Secondly, you can have another proof reading. It still has some typos such as "limited there research through" line - 833.

Author Response

Respected Reviewer,

We are very much delighted that the changes in the paper are up to your satisfaction and made you happy. We thank you for your valuable suggestion and comments which helped us to make the paper better.

Now we have again updated the paper with your suggestions and have conducted and included an in-depth discussion on the limitations, challenges, and the open future directions In the Cloud storage based healthcare domain.

We have added new sections on pages 24-28. The new section entitled are :

  • Limitations, Challenges, and Future Open Directions
  • Limitations of Healthcare Cloud Storage Domain
  • Research Challenges
  • Future Open Directions 

Along with the addition of the above sections, we have corrected and revised the paper for all grammar, typos, punctuations, structures, etc. All of these have been corrected in every section and every page. 

We Have also added and updated the latest references related to our work domain.

We hope that these changes will also satisfy you up to the mark.

Thanking for your valuable suggestions and feedback.

Reviewer 2 Report

The revised paper has addressed all my previous comments, and I suggest to ACCEPT the paper as it is now.

Author Response

Respected Reviewer,

We are very much happy and delighted that the changes in the paper are up to your satisfaction and made you happy. We thank you for your valuable suggestion and comments which helped us to make the paper better.

Thanking for your valuable suggestions and feedback.